# Tildrakizumab: Efficacy, Safety and Survival in Mid-Term (52 Weeks) in Three Tertiary Hospitals in Andalucia (Spain)

**DOI:** 10.3390/jcm11175098

**Published:** 2022-08-30

**Authors:** Ricardo Ruiz-Villaverde, Lourdes Rodriguez Fernandez-Freire, Pilar Font-Ugalde, Manuel Galan-Gutierrez

**Affiliations:** 1Department of Dermatology, Servicio de Dermatología, Hospital Universitario San Cecilio, Avenida de la Investigación s/n, 18016 Granada, Spain; 2Spain Biohealth Research Institute in Granada (ibs.GRANADA), 18012 Granada, Spain; 3Department of Dermatology, Hospital Universitario Virgen del Rocio, 41013 Sevilla, Spain; 4Department of Dermatology, Hospital Universitario San Reina Sofía, IMIBIC, 14004 Córdoba, Spain; 5Hospital Universitario San Reina Sofía, IMIBIC, 14004 Córdoba, Spain

**Keywords:** psoriasis, biological therapy, treatment

## Abstract

Tildrakizumab (TIL) binds selectively to the p19 subunit of interleukin 23. Its introduction has managed to increase the levels of efficacy, safety (improving that previously presented by the anti-IL-12/23 class) and survival. Retrospective analysis of a multicenter, observational study of real clinical practice including patients with moderate-to-severe plaque psoriasis in treatment with TIL. This cross-sectional analysis includes information of patients between February 2019 to February 2022. A total of three tertiary hospitals in Andalusia (Spain) participated in this study. Analyses were performed “as observed” using IBM SPSS v28 for Windows. A total of 61 patients were included in the analysis. The mean age of our patients was 49.5 years; 50.18% of the patients were female and 34.42% of the patients had a BMI greater than 30. It was notable that 44.26% of our patients had scalp involvement. Almost 35% of the patients had psoriatic arthropathy, although skin involvement was predominant. At week 52 (n = 34), 68% of the patients presented an absolute PASI equal to or less than 1. Regarding the drug survival, eight patients discontinued treatment due to inefficacy: five primary and three secondary failures, and one death due to causes not drug related showing survival of 86% at week 52. In the analysis of subgroups of patients, we found that scalp involvement determined greater survival (94%), as well as a shorter duration of the disease (91.7% vs. 84.4% in those with less than 10 years versus more than 15 years of evolution) and with a lower number of previous biological therapies (100% naïve, 90% in those who have used one line of biological therapy and 82.1% in those who have completed two or more lines of biological treatment. Tildrakizumab showed excellent results in the control of psoriasis in the mid-term with an elevated number of patients maintaining treatment after 52 weeks. There were no statistically significant differences in the efficiency, safety or survival results of TIL between patients coming from previous therapies.

## 1. Introduction

Tildrakizumab is an IgG1/κ type monoclonal antibody that binds specifically to the p19 subunit of the cytokine interleukin 23 (IL-23), without binding to IL-12, and inhibits its interaction with the IL-23 receptor.

In 2018, tildrakizumab was approved for the treatment of patients with moderate or severe psoriasis by the Food and Drug Administration (FDA) and the European Medicines Agency (EMA), but only the latter regulatory agency approved its use at a dose of 200 mg in patients weighing more than 90 kg [1].

The efficacy and safety of tildrakizumab have been evaluated in two randomized phase III, multicenter, double clinical trials blinded, placebo-controlled (reSURFACE 1 and 2) and etanercept (reSURFACE 2) in patients with moderate to severe plaque psoriasis who are serious candidates for systemic treatment. Both trials included a controlled treatment period of 12 weeks, a period double-blind controlled treatment up to 52/64 weeks and an extension phase. The doses of tildrakizumab for the trials of phase III were selected based on the results of the phase IIb trial, P003. Currently, the results of the extension studies of up to 148 weeks of said clinical trials have already been published [2].

Our research team previously published the results of our patient series to verify the evaluation of tildrakizumab in a short-term period [3]. The objective of this study is to evaluate the efficiency, safety and survival of tildrakizumab in the medium term (52 weeks) and compare the results with data previously published in other real clinical practice series and previous clinical trials.

## 2. Materials and Methods

### 2.1. Study Design

Retrospective analysis of a multicenter, observational study of real clinical practice including patients with moderate-to-severe plaque psoriasis in treatment with tildrakizumab. This cross-sectional analysis includes information of patients between February 2019 to February 2022. A total of three tertiary hospitals in Andalusia (Spain) participated in this study. This study has been approved by Ethics Commitee of Hospital Universitario San Cecilio (DER-HUSC-005). Before inclusion into the study, patients gave their informed consent.

### 2.2. Patients

Inclusion criteria

(1)Adult moderate-to-severe plaque psoriasis patients,(2)Psoriasis diagnosis since ≥ 1 year,(3)Patients who experimented unspecified inefficacy, primary or secondary failure to anti-TNFα, anti-IL17 and/or anti-IL12/23 as well as patients who discontinued treatment due to adverse events (AE),(4)Patients on tildakizumab treatment 100 mg s.c. (at week 0 and 4, followed by a maintenance dose every 12 weeks). None of the patients required a dose of 200 mg, which on the label can be recommended in patients weighing more than 90 kg.

Exclusion criteria were:(1)Other types of PSORIASIS different than psoriasis vulgaris (generalized or palmoplantar pustular psoriasis, erythrodermic psoriasis),(2)Presence of other inflammatory diseases such as rheumatoid arthritis, Crohn disease, ulcerative colitis and/or ankylosing spondylitis,(3)Participation at the time of data collection in a randomized clinical trial.

Missing data at different timepoints was due in part to SARS-CoV-2 pandemic, as some patients refused to visit the hospital for medical follow up.

The Psoriasis Group of the AEDV (GPs), based on the existing evidence and in coherence with the position of other national and international scientific societies (EADV-SPIN, AAD, IPC), through the Spanish Academy of Dermatology and Venereology (AEDV) issued a statement (GPs-COVID-19) in order to guide dermatologists who treat psoriasis, particularly in cases of patients undergoing treatment or who are going to start treatment with selective immunomodulatory or immunosuppressive drugs [4].

Available data on past and present outbreaks of Coronavirus infections (SARS, MERS, COVID-19) suggest that immunosuppressed patients are not at increased risk of severe manifestations and complications of COVID-19 compared to the general population.

Our performance in real clinical practice followed three of the points mentioned in this position paper:(a)Psoriasis patients can, in general, continue their treatment during the COVID-19 pandemic in order to avoid psoriasis outbreaks that would directly impact the patient and the use of greater healthcare resources.(b)In general, it is recommended to discontinue small molecule immunosuppressive therapy or biological therapy in patients with suspected active COVID-19 infection. Starting these treatments in case of active infection would be contraindicated.(c)If patients live in areas with a high incidence of COVID-19 infection, or in close contact with confirmed cases, the temporary interruption of some selective immunosuppressive or immunomodulatory therapies should be evaluated considering factors such as age or comorbidities until confirming or ruling out infection.

In short, some of the patients only missed one face-to-face visit between March 2020 and June 2020. The rest of the visits have always been face-to-face. On the other hand, there have been no significant treatment interruptions except in cases of active infection, where the patient suspended treatment if the PCR-COVID-19 was positive until it became negative. We cannot establish a comparison with the efficacy of the drug in the pre-pandemic period, as it was marketed a few months prior to its onset.

### 2.3. Treatment

Patients received tildrakizumab following data sheet specifications (100 mg administered subcutaneously at week 0, 4 and followed by a maintenance every 12 weeks). The treatment began in an underhanded manner due to the failure of the previous treatment without a bleaching period.

### 2.4. Outcome Measures

Disease severity and treatment response was assessed by absolute psoriasis area and severity index (PASI), body surface area (BSA), VAS pruritus, dermatology life quality index (DLQI) and PASI < 3, PASI < 2 and PASI < 1 at 0, 12, 24 and 52 weeks.

### 2.5. Safety

Safety and tolerability to tildrakizumab were evaluated during the follow-up of the study (any adverse event experienced by the patient was reported). The analysis included discontinuations due to lack of effectivity and safety reasons. Primary failure was considered failure to reach PASI 90 after applying the biologics and secondary failure is defined as failure to maintain PASI 90 after 12 weeks of treatment. No laboratory testing was assessed.

### 2.6. Statistical Analysis

A descriptive analysis of the evaluated variables was carried out. Means and standard deviations were calculated for quantitative variables and absolute values and percentages for categorical variables. Means were compared using the Student’s *t*-test.

Survival was estimated by Kaplan–Meier curves. Any reason for discontinuation was reported and used for the analysis. Treatment survival was calculated using Kaplan–Meier survival analysis. *p* values < 0.05 were considered statistically significant. Analyses were performed “as observed” using IBM SPSS v28 for Windows^®^.

## 3. Results

Our study included 61 patients with moderate–severe psoriasis treated with tildrakizumab, who had completed at least 12 weeks of treatment. The baseline characteristics of the sample are reported in Table 1. The mean age of our patients was 49.5 years. The mean number of years of evolution of the disease was 19.5 years. Finally, 34.2% of the patients had a BMI > 30 (obesity).

Overall, 44.26% of the patients had significant involvement of the scalp. Among the comorbidities analyzed, 34.42% of the patients presented psoriatic (peripheral) arthropathy. In our cohort, five patients had developed solid neoplasia, for which they were currently controlled.

Only 4.91% of the patients were biological naïve and 49.18% of our patients had previously received at least one biological drug.

Of the 61 patients included in our study, 34 of them reached week 52. Of these, five patients abandoned treatment due to primary failure, two due to secondary failure, one was lost to follow-up, and one died due to causes unrelated to the medication in question. No minor or serious adverse effects were reported during treatment with tildrakizumab and no patient discontinued treatment for safety reasons.

In relation to the main measures of efficacy, the evolution of the percentage of patients who reached absolute PASI < 3, <2, <1 or equal to 0 are reflected in Table 2.

In the bivariate analysis carried out by subgroups, we did not find statistically significant differences (*p* < 0.05) between bionaïve or bioexperienced patients, patients with BMI < 30 or BMI > 30 in relation to the efficacy variables (PASI and BSA) and analyzed PROs (VAS pruritus and DLQI). We did not find statistically significant differences between genders either, and scalp, nail or reverse involvement were not associated with the presence of psoriatic arthropathy.

Overall drug survival at week 52 was 86%. In the analysis of different subgroups of patients, we were able to find that the involvement of the scalp determined a greater survival of the drug concerning other locations (94%). The shorter duration of the disease is associated with longer survival of the drug, similar to what has been published with other molecules (91.7% vs. 84.4% in those with less than 10 years of evolution vs. more than 15 years). Finally, the number of previous biological therapies also influenced the observed survival of the drug at 52 weeks (100% in naïve, 90% in those who have used one line of biological therapy, and 82.1% in those who have completed two or more biological treatment lines) (Figure 1A–D).

## 4. Discussion

On randomized clinical trials, the inclusion and exclusion criteria must be strict and, therefore, patient profiles that are found in our real clinical practice are not considered. For this reason, it is important to compare the results of these trials with the efficacy, safety and survival data at different time cut-off points: short term (12–16w), mid-term (52w) in which the robustness of the drug is observed and long-term (>104w).

There are different series of real clinical practice that have been published in relation to the management of patients with moderate–severe psoriasis with tildrakizumab. Burlando et al. [5] (n = 26; 24 weeks), Wei et al. [1] (n = 30; 52 weeks); Drerup [6] (n = 150; n = 58 at 76 weeks); Caldarola [7] et al. (n = 55; 28 weeks).

Regarding the mean age of the patients included, the Wei series stands out [1], with a mean age of 60.6 years-old, which curiously is the one with the lowest effectiveness rates compared to the rest of the clinical real practice series and clinical trials, attributing it to the comorbidities that patients of this age group present.

There are no significant differences in terms of BMI, although in our series the percentage of obese patients is significantly higher.

In our opinion, a particularly important consideration is the percentage of bio-naïve patients, since in our case it is only 5%, while in the Caldarola series [6] it reaches 60%, without showing any special differences in the results of efficiency achieved. This data is due to the fact that, in our region, the local regulatory agency strongly recommends that all patients with moderate–severe bio-naïve psoriasis should go through a biosimilar TNF blocker drug with exceptions linked to the patient’s severity status.

The aggregate analysis of efficacy data at 52 weeks from the reSURFACE 1 and 2 clinical trials shows that 72% and 43% of patients maintained PASI 90 and PASI 100 with the 100 mg tildrakizumab dose [2]. These data are consistent with those published by Wei [1] (60% PASI90; 33.3% PASI100) and Drerup [6] (71% PASI < 3; 39% PASI < 1). However, our data improve both the data of the clinical trials and those of the previously mentioned series (91% PASI < 3; 68% PASI < 1).

Blauvet et al. [8] carry out an analysis of the efficacy and impact on quality of life up to 52 weeks in patients as a result of a pooled analysis of two randomized controlled trials. Temporarily, the results can be compared with our series despite a larger sample size (n = 575 patients) and the use of the two doses allowed by the data sheet. One of the main conclusions established by the authors is that the differences in responses between patients who eventually achieved PASI improvement < 50% and ≥50% could be differentiated as early as week 8 (i.e., after two doses of tildrakizumab). Although the efficacy endpoints used differ in our analysis (PASI absolute) from the Blauvelt study (PASI 75,90,100), the trend and morphology of the curves are analogous. PASI improvement from baseline at week 28 appeared to have continued improvement beyond week 28. On the other hand, it is confirmed that a better skin clearance was associated in both studies with a greater likelihood of patients achieving DLQI 0/1 over 52 weeks.

Another piece of information of interest in our series is that five patients had developed a previous solid neoplasm, for which they were controlled. This patient profile has only been previously reported by Wei et al. [1] (n = 9, 30% of patients) since history of previous malignancy is an exclusion criterion in the reSURFACE studies. It is a finding that, from our point of view, enhances the safety of the drug since, even in this population at risk, the development of new neoplasms or reactivation of previous ones have not been observed in a follow-up period of 12 months.

On the other hand, in the survival analysis, we were able to observe how certain factors influence a greater persistence of the treatment, such as a shorter duration of the disease or the use of a smaller number of previous biological lines of treatment. We have observed that patients with scalp involvement showed greater survival of the drug, suggesting the possibility that, when it exists, it is a particularly interesting therapeutic option.

Of special interest is the study by Thaci et al. [9]. In this study the proportions of tildrakizumab 100 mg responders achieved in absolute PASI scores < 5/<3/<1 were 96.4%, 85.1% and 50.8% at week 28 and 88.7%, 78.8% and 47.7% at week 244. Our results at week 52 are in line with those reported previously. Although in this sub-analysis, unlike our study, the survival of the drug is not evaluated, it is possible to observe a certain parallelism in the efficacy and survival data in relation to withdrawals due to adverse events and secondary failures. The main limitation of our study is the retrospective nature of the analysis, the loss of some data during follow-up as a result of the SARS-CoV-2 pandemic, and the sample size, although we believe that it is representative and comparable to that of other published series. Although there is concern that drugs used in treatment of some patients with psoriasis may increase the risk and severity of infection with COVID-19, we agree with Ebrahimi [10] that while there is not definitive controlled trial data, the available evidence suggests that patients with psoriasis without COVID-19 can continue the biological therapy for psoriasis. Furthermore, TNFα inhibitors may even be protective of severe COVID-19 relative to other treatments or no treatment at all, and biological treatment does not seem to have a significant impact on the response and safety of vaccines in patients with psoriasis treated with biologicals [11,12].

The ultimate goal of moderate–severe psoriasis treatment is to achieve complete whitening of skin symptoms, both in the short term and long term. The treatment selection should take into account the characteristics of the disease (severity, location, joint involvement, degree of activity of the disease) as well as factors related to the patient (age, previous treatments, comorbidities, expectations and lifestyle) and to the treatment itself (efficacy and safety). In conclusion, our experience confirms the efficacy, safety and survival of tildrakizumab in real-life practice in follow-up periods of 52 weeks, with baseline characteristics of our patients with a worse prognosis than in other previously published series. Further research is needed to demonstrate the survival of the drug beyond 104 weeks in order to verify if the consistency demonstrated in the sub-analyses of the clinical trials is maintained in real clinical practice.

## Figures and Tables

**Figure 1 jcm-11-05098-f001:**
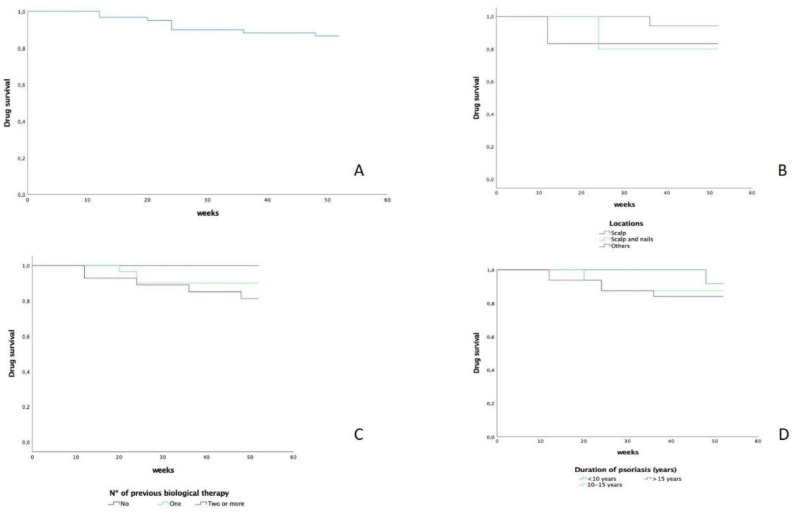
(**A**). Survival of tildrakizumab in a 52w period. (**B**). Survival regarding locations of psoriasis. (**C**). Survival regarding previous lines of treatment. (**D**). Survival regarding time duration of psoriasis.

**Table 1 jcm-11-05098-t001:** Baseline characteristics of our series.

Basal Data	SD	N = 61
**Age (years), media (SD)**	13.90	49.5
**Sex, n (%)**		
Male		30 (49.18%)
Women		31 (50.81%)
**Disease evolution (years), media (SD)**	9.76	19.24
**BMI**		29.01
Underweight		0
Normal weight (BMI 18.5–24.9)		17 (27.86%)
Overweight (BMI 25–29.9)		23 (37.70%)
Obesity (BMI ≥ 30)		21 (34.42%)
**Psoriasis type, n (%)**		
Plaque		59 (96.72%)
Nails		12 (19.67%)
Scalp		27 (44.26%)
Palmoplantar		3 (4.91%)
Inversa		2 (3.27%)
**Comorbidities**		
Psoriatic arthritis		21 (34.42%)
Metabolic syndrome		12 (19.67%)
Diabetes		8 (13.11%)
Hypertension		15 (24.59%)
Dyslipidemia		21 (34.42%)
Psychiatric comorbidity		6 (9.83%)
Non-alcoholic fatty liver		5 (8.19%)
**PASI, media**	5.5	10.65
**BSA, media**	8.3	14.45
**VAS pruritus, media**		
**PGA, medio**	1.2	3.37
**DLQI, media**	6.5	15.75
**Number of previous biological treatments**		
0		3 (4.91%)
1		30 (49.18%)
2		13 (21.31%)
3		10 (16.39%)
4		3 (4.91%)
5		2 (3.27%)

**Table 2 jcm-11-05098-t002:** Evolution of PASI, BSA andDLQI at the different time cut-off points.

	Basal (n = 61)	12w (n = 55)	24w (n = 47)	52w (n = 34)
**PASI, media**	10.65	2.94	1.88	1.38
**PASI reduction**				87%
**PASI ≤ 3**		76%	81%	91%
**PASI ≤ 2**		67%	74%	85%
**PASI ≤ 1**		45%	57%	68%
**BSA, media**	14.45	3.41	1.93	1.64
**BSA reduction**				89%
**PGA, media**	3.37	1.24	1.24	1.05
**PGA reduction**				69%
**DLQI, media**	15.75	3.38	3.07	2.24
**DLQI reduction**				86%

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
