# Peer review of "Tildrakizumab: Efficacy, Safety and Survival in Mid-Term (52 Weeks) in Three Tertiary Hospitals in Andalucia (Spain)"

_jcm, 2022, doi:10.3390/jcm11175098_

Round 1

Reviewer 1 Report

1.The work required additional scientific clarity, as summarized in the comments below:

The authors Ricardo Ruiz-Villaverde et al., addressed Tildrakizumab:
Efficacy, safety, and survival in mid-term (52 weeks) on real clinical
practice,

I feel the title should be modified to explain the actual work in which
a total of three tertiary hospitals in Andalusia (Spain) participated in
this study, but the title explains real clinical practice instead they
should mention as “region-specific” which is opt at this context.

2. Another concern is what is the difference from the previously
published

a) Blauvelt, A., et al. "Tildrakizumab efficacy and impact on quality of life up to 52 weeks in patients with moderate‐to‐severe psoriasis: a pooled analysis of two randomized controlled trials."
Journal of the European Academy of Dermatology and Venereology 33.12
(2019): 2305-2312.

b) Thaci, D., et al. "Five‐year efficacy and safety of tildrakizumab
in patients with moderate‐to‐severe psoriasis who respond at week 28:
pooled analyses of two randomized phase III clinical trials (reSURFACE 1
and reSURFACE 2)." British Journal of Dermatology 185.2 (2021): 323-334.

Perhaps rather than the region-specific.

3.According to the study, 50.18% of the patients were female, so we were very curious about any gender-related differences that can be expected, if it is such cases why authors did not place 50% female and 50% male populations?

4. As I realized and authors themselves claimed the limitation of these studies performed on the information of patients between Feb 2019 to Feb 2022, as we know that during the study time, COVID-19 was first reported in late Dec 2019 and, since then, has rapidly taken over the globe, still COVID-19 crisis continues around the globe.

COVID-19 patients and psoriasis patients will have similar dysregulationof the immune system, and the immunological abnormalities in COVID-19 patients can also be observed in auto-inflammatory or autoimmune conditions “psoriasis”.

Therefore, how this efficacy and other
mechanistic things could be compared with the non-COVID-19 era ie.
Pre-COVID-19 and during COVID-19 and post-COVID-19, how the data can be
compared? The authors should clarify

(a) The impact of psoriasis on COVID-19, and vice versa, and

(b) The impact of COVID-19 on the selection of psoriasis therapies

5. The authors conclude that Further research is needed to demonstrate the survival of the drug beyond 104 weeks. We are very curious about whether the work continues as similar to the previously Thaci, D., et al. "Five‐year” reported studies.

Author Response

We would like to thank the reviewer te criticism perfomed

The authors Ricardo Ruiz-Villaverde et al., addressed Tildrakizumab:
Efficacy, safety, and survival in mid-term (52 weeks) on real clinical
practice,

I feel the title should be modified to explain the actual work in which
a total of three tertiary hospitals in Andalusia (Spain) participated in
this study, but the title explains real clinical practice instead they
should mention as “region-specific” which is opt at this context.

AUTHOR REPLY: We have proceeded to change the title in accordance with the suggestions of the reviewer

2. Another concern is what is the difference from the previously
published

a) Blauvelt, A., et al. "Tildrakizumab efficacy and impact on quality of life up to 52 weeks in patients with moderate‐to‐severe psoriasis: a pooled analysis of two randomized controlled trials."
Journal of the European Academy of Dermatology and Venereology 33.12
(2019): 2305-2312.

b) Thaci, D., et al. "Five‐year efficacy and safety of tildrakizumab
in patients with moderate‐to‐severe psoriasis who respond at week 28:
pooled analyses of two randomized phase III clinical trials (reSURFACE 1
and reSURFACE 2)." British Journal of Dermatology 185.2 (2021): 323-334.

Perhaps rather than the region-specific.

AUTHOR REPLY: We have proceeded to carry out in the Discussion section an extension of it with the results of the two indicated studies that we had not initially considered and that we think are of the greatest interest by adding data from randomized clinical trials.

3.According to the study, 50.18% of the patients were female, so we were very curious about any gender-related differences that can be expected, if it is such cases why authors did not place 50% female and 50% male populations?

AUTHOR REPLY: We have not found statistically significant differences in the bivariate analysis carried out, but we consider that it is a parity sample in terms of gender. The aim of the present study was not to perform a sub-analysis of these characteristics.

4. As I realized and authors themselves claimed the limitation of these studies performed on the information of patients between Feb 2019 to Feb 2022, as we know that during the study time, COVID-19 was first reported in late Dec 2019 and, since then, has rapidly taken over the globe, still COVID-19 crisis continues around the globe. COVID-19 patients and psoriasis patients will have similar dysregulationof the immune system, and the immunological abnormalities in COVID-19 patients can also be observed in auto-inflammatory or autoimmune conditions “psoriasis”.

 Therefore, how this efficacy and other mechanistic things could be compared with the non-COVID-19 era ie. Pre-COVID-19 and during COVID-19 and post-COVID-19, how the data can be compared? The authors should clarify

(a) The impact of psoriasis on COVID-19, and vice versa, and

(b) The impact of COVID-19 on the selection of psoriasis therapies

AUTHOR REPLY: Thank you very much for your consideration. We have expanded how we carry out reviews in the first wave of the COVID-19 pandemic, by following the AEDV position paper. Basically, some of the patients only missed one face-to-face visit between March 2020 and June 2020. The rest of the visits have always been face-to-face. On the other hand, there have been no significant treatment interruptions except in cases of active infection, where the patient suspended treatment if the PCR was positive until it became negative. We cannot establish a comparison with the efficacy of the drug in the pre-pandemic period, as it was marketed a few months prior to its onset.

5. The authors conclude that Further research is needed to demonstrate the survival of the drug beyond 104 weeks. We are very curious about whether the work continues as similar to the previously Thaci, D., et al. "Five‐year” reported studies.

AUTHOR REPLY: Thanks for your comment. The initial idea is once the efficacy and safety data have been examined in the short term (16w) and medium term (52w), see the survival of the drug in the long term (2yr or 5yr), since we have had the experience with other anti-IL23 biological drugs (i.e Guselkumab) to maintain long-term survival.

The new version of the manuscript includes all the changes in a track changes version so that the reviewer can see the improvements we have made as a result of their comments.

Reviewer 2 Report

Authors present their experience on Tildrakizumab in a real life environment. Their article is logically presented and sounded. 

I have the following comments:

Did you find any relationship between those who had scalp involvement and arthropathy?

Table 1 

Tipo de psoriasis and comorbilidades should be translated

Table 2 does not show any data regarding VAS pruritus change. 

Author Response

Thank you very much for your kind review. We come to comment on your reviews

1. Did you find any relationship between those who had scalp involvement and arthropathy? AUTHOR REPLY: In the bivariate analysis we have not found any relationship between the presence of psoriatic arthritis and scalp involvement, despite the fact that in many articles it is indicated as a predictor of joint involvement.

2. Table 1 Tipo de psoriasis and comorbilidades should be translated. AUTHOR REPLY: It has been traslated

3. Table 2 does not show any data regarding VAS pruritus change. AUTHOR REPLY: It has been reviewed

Regards

Round 2

Reviewer 2 Report

Thank you for the changes. I don't have further comments.